

# Initial Results From a Field Campaign of Wake Steering Applied at a Commercial Wind Farm: Part 1

Paul Fleming[1], Jennifer King[1], Katherine Dykes[1], Eric Simley[1], Jason Roadman[1], Andrew Scholbrock[1], Patrick Murphy[1,3], Julie K. Lundquist[1,3], Patrick Moriarty[1], Katherine Fleming[1], Jeroen van Dam[1], Christopher Bay[1], Rafael Mudafort[1], Hector Lopez[2], Jason Skopek[2], Michael Scott[2], Brady Ryan[2], Charles Guernsey[2], and Dan Brake[2]

[1]National Wind Technology Center, National Renewable Energy Laboratory, Golden, CO, 80401, USA
[2]NextEra Energy Resources, 700 Universe Blvd, Juno Beach, FL 33408
[3]Dept. Atmospheric and Oceanic Sciences, University of Colorado Boulder, Boulder, CO, 80303, USA

*Correspondence to:* Paul Fleming (paul.fleming@nrel.gov)

**Abstract.** Wake steering is a form of wind farm control in which turbines use yaw offsets to affect wakes in order to yield an increase in total energy production. In this first phase of a study of wake steering at a commercial wind farm, two turbines implement a schedule of offsets. Results exploring the observed performance of wake steering are presented, as well as some first lessons learned. For two closely spaced turbines, an approximate 13% increase in energy was measured on the downstream turbine over a 10° sector. Additionally, the increase of energy for the combined upstream/downstream pair was found to be in-line with prior predictions. Finally, the influence of atmospheric stability over the results is explored.

### Copyright Statement

This work was authored by the National Renewable Energy Laboratory, operated by Alliance for Sustainable Energy, LLC, for the U.S. Department of Energy (DOE) under Contract No. DE-AC36-08GO28308. Funding provided by the U.S. Department of Energy Office of Energy Efficiency and Renewable Energy Wind Energy Technologies Office. The views expressed in the article do not necessarily represent the views of the DOE or the U.S. Government. The U.S. Government retains and the publisher, by accepting the article for publication, acknowledges that the U.S. Government retains a nonexclusive, paid-up, irrevocable, worldwide license to publish or reproduce the published form of this work, or allow others to do so, for U.S. Government purposes.

## 1 Introduction

Wind farm control is a field of research in which the control actions of individual turbines are coordinated to improve the total performance of the wind farm as defined by the total power production of the wind farm as well as the loads experienced by downwind turbines. Wake steering is a form of wind farm control wherein an upstream turbine intentionally offsets its yaw angle with respect to the wind direction to benefit downstream turbines. Wagenaar et al. (2012); Dahlberg and Medici (2003).





Wake steering has been studied through wind tunnel studies [e.g., Medici and Alfredsson (2006); Park et al. (2016); Schottler et al. (2017); Bartl et al. (2018)], and large-eddy simulation (LES) studies of wake steering have been undertaken to date [see, for example, Fleming et al. (2015); Vollmer et al. (2016); Howland et al. (2016)]. Coupled with theoretical derivations, the results of the previously mentioned studies have enabled the development of control-oriented engineering models of wakes

and wake steering that can be used to design and analyze wake steering controllers for wind farms, as well as predict the performance benefit. Important examples include the Jensen wake model (Jensen (1984)) and the model of wake steering of Jiménez et al. (2010).

FLOw Redirection and Induction in Steady State (FLORIS) is a software repository that provides an engineering model of wake steering that can be used in the design and analysis of wind farm control applications (Gebraad et al. (2016)). Originally

based on Jensen (1984) and Jiménez et al. (2010), it now employs the wake recovery and redirection models of Bastankhah and Porté-Agel (2014, 2016); Niayifar and Porté-Agel (2015). Annoni et al. (2018) provide a detailed description of the current FLORIS model. The FLORIS model is open source and available for download and collaborative development (https://github.com/WISDEM/FLORIS). Development is ongoing, and future models will incorporate the advances proposed in Martínez-Tossas et al. (2018).

Critical to advancement, improvement, and eventual adoption of wake steering are field trials of wake steering in realistic environments. Wagenaar et al. (2012) attempted wake steering at a scaled wind farm. One important campaign took place at the National Wind Technology Center in Boulder, Colorado. In that study, a rear-facing scanning lidar from the Univesity of Stuttgart was placed on top of the nacelle of a GE 1.5-MW turbine, which held various yaw offset positions for periods of one hour at a time (Fleming et al. (2017a)). The data from that campaign were used to investigate the accuracy of predictions made

by FLORIS (Annoni et al. (2018)) as well as the impact on turbine loads caused by yaw offsets (Damiani et al. (2017)). A related campaign is being undertaken at the Scaled Wind Farm Technology facility in Lubbock, Texas. Similar to the National Wind Technology Center study, a rear-facing lidar (in this case the Technical University of Denmark spinner lidar) is used to scan the wake of a V27 experimental turbine. The resulting data are used to examine wake behavior (Herges et al. (2017)) as well as understand loading impacts (White et al. (2018)). Finally, a first published field trial at a commercial offshore wind

farm is presented in Fleming et al. (2017b). In that study, a single turbine implements a yaw-offset control strategy to benefit three downstream turbines.

Still, there is a need for more conclusive field campaigns on the performance of wake steering, as well as evaluation of the latest models. For this reason, a new field campaign was initiated as a collaboration between the National Renewable Energy Laboratory (NREL) and NextEra Energy Resources. A portion of a commercial wind farm was selected as a test site, and

significant additional sensing equipment is being deployed including a (ground-based) lidar, meteorological (met) tower, and two sodars. Additional nacelle-based lidars are being deployed for the upcoming second phase. Wake steering controls based on the latest version of FLORIS are implemented on two turbines. This paper presents the results of the first phase of this campaign focused on wake steering.

The main contribution of this paper is the initial results and analysis of a land-based, wake steering field-test campaign. The

paper presents the controller as implemented in the present phase and proposes improvements based on these initial results.





The performance of wake steering, in terms of increased energy production, is analyzed and compared with predictions from the FLORIS code. In addition, the wake steering performance is assessed with respect to atmospheric stability, which can be estimated using sensing available on the met mast. Finally, several practical lessons learned are discussed.

The paper is organized as follows. Section 2 provides an overview of the field campaign's layout of turbines and sensors, as well as meterological conditions. Section 3 discusses the implemented controller. Section 4 describes the data collected, in terms of total amount and characteristics. The performance of the controller, specifically in terms of achieving targeted offsets, is reviewed in Section 5. Challenges specific to this first phase are described in Section 6. Finally, Section 7 presents the results.

## 2  Field Campaign

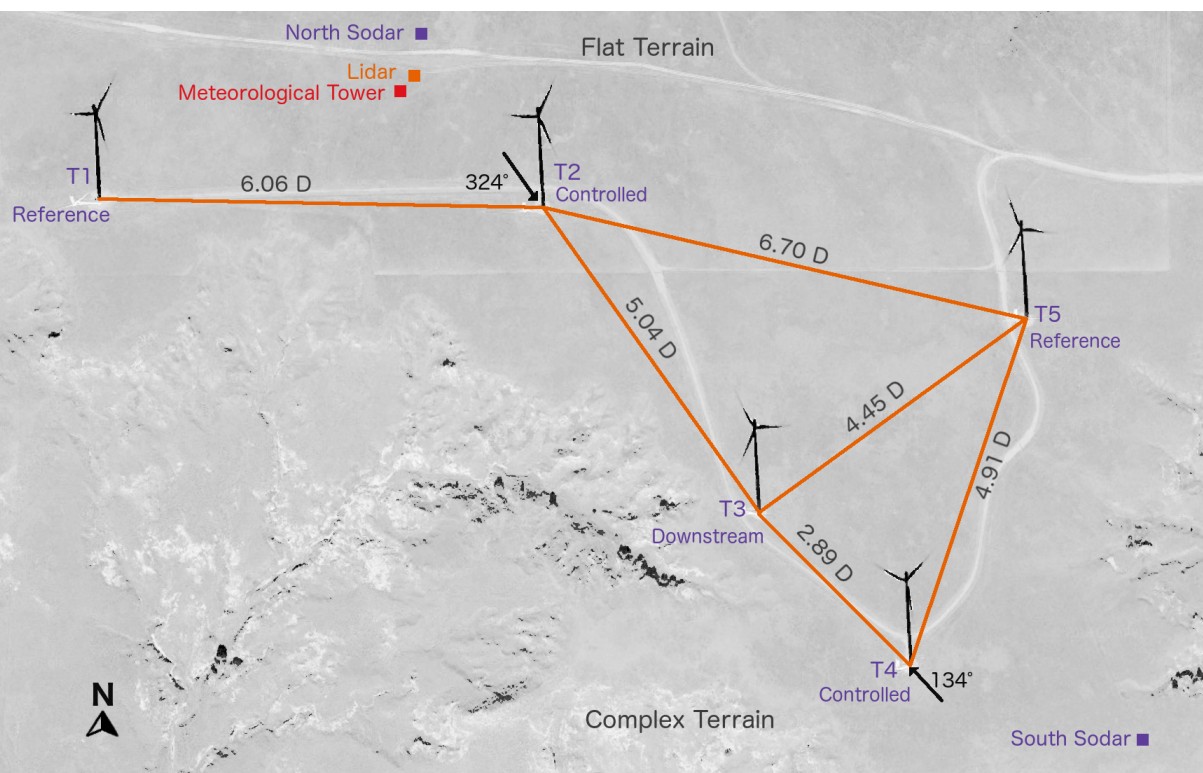

**Figure 1.** Layout of the experimental site. Turbines 2 (T2) and Turbine 4 (T4) have wake steering implemented to benefit Turbine 3 (T3), whereas Turbine 1 (T1) and Turbine 5 (T5) are reference turbines. The position of the installed meteorological equipment is also shown. Finally, the complexity of the terrain to the south and flat terrain to the north are indicated.

A subsection of a commercial wind farm was selected as the test site for the wake steering campaign. The site was chosen to include a set of turbines where the main wind directions that generate strong waking conditions would occur relatively





frequently and the turbines were close enough for wake steering effects to be discernible. The selected wind farm subsection is shown in Fig. 1.

Five turbines (Fig. 1) are located in one corner of the overall farm. Note there are no turbines to the north or south, making these wind directions effectively freestream. The five turbines are relatively closely spaced, especially the three turbines labeled

T2, T3, and T4. T2 and T4 were controlled turbines, and T3 was selected as the downstream turbine to be evaluated based on wake impacts of T2 and T4 . The wind directions that T2 (324°) or T4 (134°) directly wake T3 are indicated in Fig. 1. T1 and T5 serve as reference turbines, uncontrolled and unaffected by the control turbines during wind conditions during which controls would be applied. Fig. 2 illustrates the directional conventions for steering applied to the T4 and T3 turbine pair.

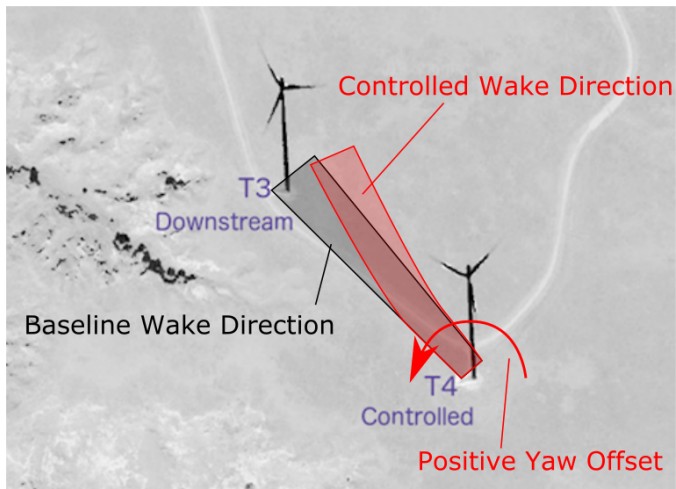

**Figure 2.** Illustration of wake steering showing that a positive yaw offset is meant to indicate a counter-clockwise rotation of the controlled turbine.

The terrain of the site is also illustrated in Fig. 1. Generally, the terrain to the north is flat, whereas the terrain to the south is

complex (some escarpments can be seen in the southwest Fig. 1 and these extend to the south of T4). The campaign is divided into the "north" campaign, where flows from the north arrive over flat terrain, and T2 is the controlled turbine, and the "south" campaign, where flows from the south arrive over complex terrain and are expected to be more turbulent.

The locations of the meteorological equipment are indicated in Fig. 1. Based on the simpler terrain and overall wind rose, the equipment is placed to prioritize the north campaign. A Leosphere Windcube v2 profiling lidar (shown in Fig. 1) provides

profiles of wind speed and wind direction calculated nominally every second but averaged to 1-min intervals. This lidar (similar to that used in Lundquist et al. (2017)) samples line-of-sight velocities in four cardinal directions along a nominally 28° azimuth from vertical, followed by a fifth vertically pointed beam. Range gates were centered every 20 meters from 40 m up to 180 m. The sodars used in the campaign are Vaisala Triton Wind Profilers. The sodars provide measurements of wind speed and wind direction every 20 m up to 200 m.





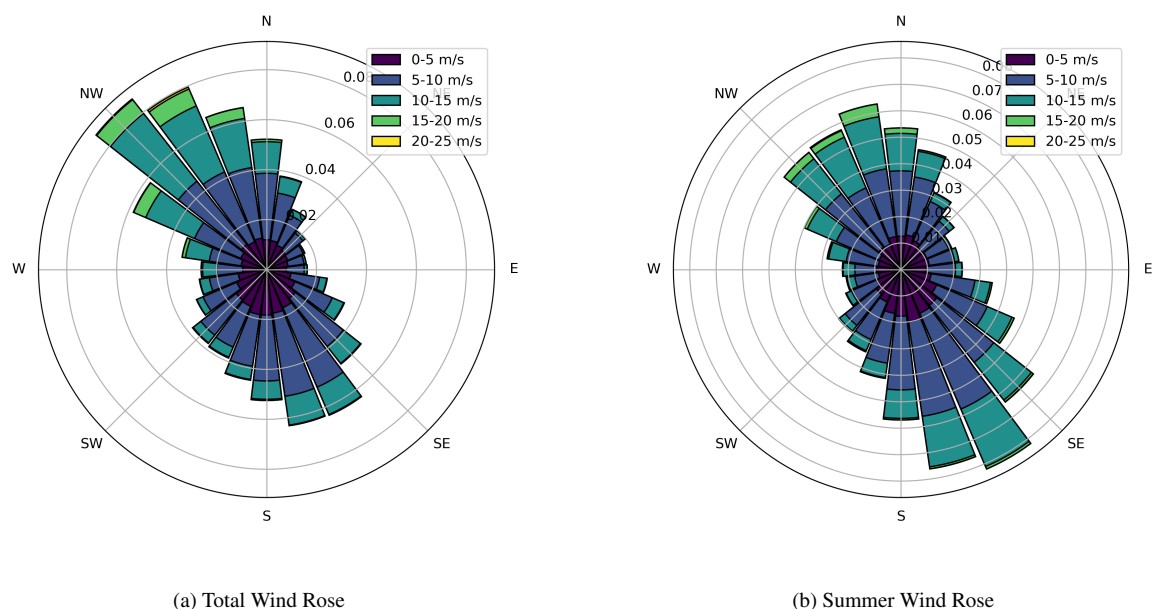

(a) Total Wind Rose

(b) Summer Wind Rose

**Figure 3.** Total wind rose for the site and for the particular months encompassing the test, based on data collected from https://www.nrel.gov/grid/wind-toolkit.html

Phase 1 of the field campaign uses an initial deployment of the wake steering controller, and initial collection of data over the summer of 2018 (from May 4 through July 11, 2018). The wind resource is seasonal: in the summer, southern winds are more probable than northern winds. Fig. 3 shows two wind roses for the site, the data for which were obtained using NREL's Wind Integration National Dataset toolkit and are for 100 m height. Fig. 3a shows the annual wind rose, with winds coming

5  dominantly from the north-northwest and south-southeast. Fig. 3b shows the expected wind rose for the months during which the campaign was run has more frequent south-southeasterly winds.

Controllers were implemented and running on both T2 and T4. Because the south-southeasterly winds are more prominent in this season, phase 1 focuses on the south campaign as most of the collected data correspond to this direction. The final study will consider the north experiment as well. Because of this focus on the south campaign, the most relevant components of

10  Fig. 1 are T4 (controlled turbine), T3 (downstream turbine), and the "south sodar," to measure inflow.

## 3  Controller

The controller implemented onto T4 was designed by optimizing a FLORIS model of the site based on wind direction and wind speed. This resulted in a look-up table, which provides a desired yaw offset for T4 as a function of wind speed and direction. Fig. 4 shows this target offset function, as a function of wind direction for the case of 8 m/s winds. The magenta lines indicate



the approximate boundary of control and will be reused in upcoming figures to distinguish controlled and uncontrolled wind directions. The offset is largest around the peak wake loss direction near 134° and decreases as the wind rotates southerly.

These offset tables were constrained to be below load impact limitations determined by Damiani et al. (2017). A safe load envelope was determined to be yaw offset angles no larger than 20° during wind speeds of 12 m/s and less. Finally, offsets

5 were restricted to be in the counterclockwise direction with respect to the wind (when viewed from above).

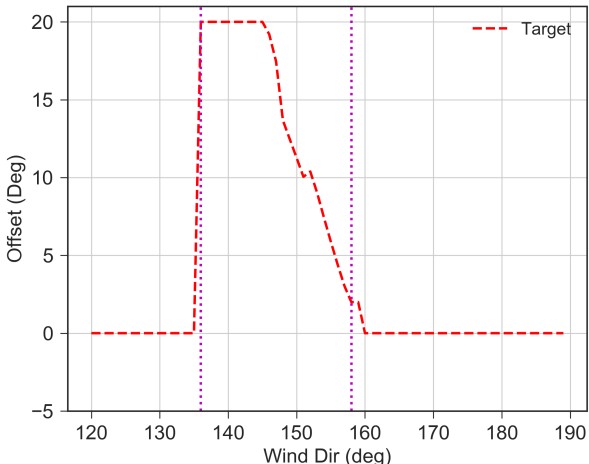

**Figure 4.** Target offset for 8 m/s. The red line indicates the target offset by wind direction. The magenta vertical lines indicate the approximate boundaries of the experiment (these vary slightly by wind speed, but the overall shape is the same).

The yaw controllers of the controlled turbines were then modified to implement this yaw offset strategy. Specifically, the nacelle vane signal fed into the controller was modified by the specified yaw offset amount in the look-up table to induce the yaw controller to track an offset. An external wake steering controller was implemented to determine the offset to apply at a given moment. The specific setup is shown in Fig. 5.

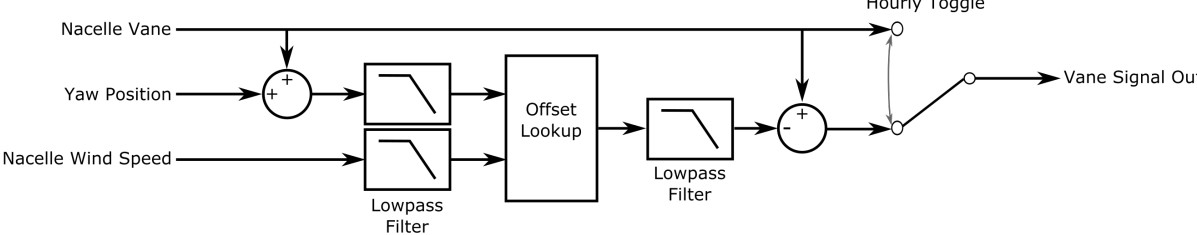

**Figure 5.** Block diagram of wake steering controller implementation. Inputs are provided by a typical turbine sensor, and the output is a modified vane signal to supply the turbine yaw controller for wake steering. Note this output is toggled on/off hourly.





The controller in Fig. 5 computes a wind speed and wind direction from sensors available on each turbine. It then filters both of these signals to remove high-frequency changes. The signals are fed into a look-up table, which is also filtered, and then the modified offset vane signal is sent to the turbine yaw controller.

The offset function is toggled on and off every hour, as indicated in the diagram. This toggling enables the performance of

the wake steering controller to be compared to a baseline control data set that includes a similar composition of wind speeds. The decision to toggle every hour was a balance between accounting for the slowness of most yaw controllers and the variation of wind conditions. The optimal toggling period should be studied in more detail in the future to optimize the usefulness of the data collected.

## 4   Data Collection

The phase 1 campaign lasted for approximately 3 months during which the wake steering offset controller on T4 was toggled on and off hourly. This section describes the inflow conditions during this period.

The inflow conditions are described from the south sodar data. Wind speed and wind direction are computed by a weighted average of the sodar measurements at heights that are within the rotor area of the turbine similar to the rotor-equivalent wind speed (Wagner et al. (2014)). Turbulence intensity (TI) is estimated at hub height by the sodar as the 10-minute standard

deviation of the wind speed divided by the mean wind speed. Finally, stability is quantified via the Obukhov lengths (Stull (2012)). For this case Obukhov length $L$ is computed via:

$$L = \frac{-u_*^3 \overline{\theta_v}}{K g \overline{w'\theta_v'}} \tag{1}$$

$$u_* = |\overline{u'w'}^2 + \overline{v'w'}^2|^{1/4} \tag{2}$$

where $K$ is the von Kármán constant assumed to be 0.4, $g$ is gravity, $\theta_v$ is virtual potential temperature calculated with the

met tower pressure at 2.5 m, and $u$,$v$, and $w$ are meteorological coordinates of wind speed components in the west-east, south-north, and vertical planes. Fluxes were calculated using a Reynolds decomposition based on a 30-minute average. Using the classification scheme of Wharton and Lundquist (2012), the data were divided into stable, neutral, and unstable conditions. This division is simplified here so that "stable" is defined as $L < -1000$ and all other data are categorized as "not stable."

The total amount of data collected is summarized in Fig. 6. The amount of data collected between the "Baseline" set,

i.e., controller off, and "Controlled," i.e., controller on, is comparable. The data are broken into "stable" and "not-stable" atmospheric conditions to show that toggling ensures both sets are similarly composed of atmospheric conditions.

Atmospheric conditions are described in Fig. 7. In Fig. 7a, the wind directions observed over the course of the campaign (as measured by the south sodar at hub height) are illustrated. If data are restricted to those points occurring in the range where the experimental controller is active (in terms of wind speed and direction), and further limited by removal of fault-coded data





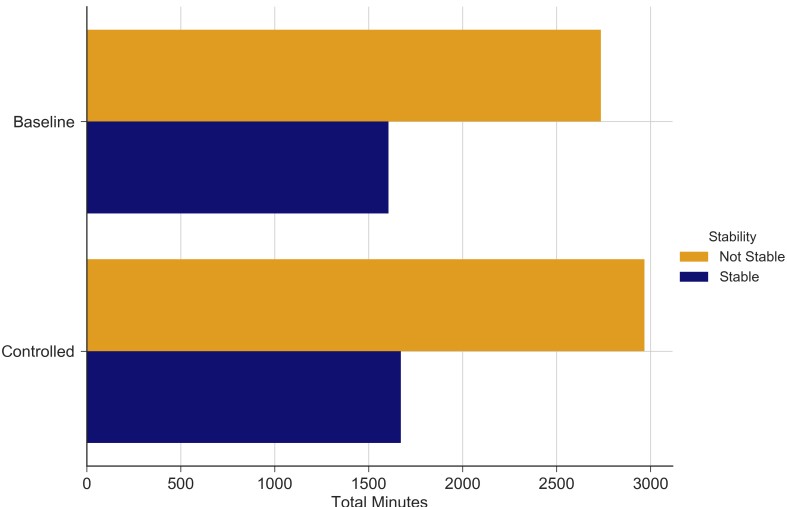

**Figure 6.** Total data collection. The yellow bars indicate the data from "not-stable" conditions and the blue bars indicate the data from "stable" conditions. The data were collected for both "Baseline" and "Controlled" cases.

or faulty sensing, the data reduce to Fig. 7b. The distribution of wind speeds making up Fig. 7b are then shown in Fig. 7c. Finally, Fig. 7d illustrates the recorded TI within the data set remaining in Fig. 7b. The box sizes in Fig. 7d indicate the amount of data, and the data are subdivided into stable and not-stable categories to show that lower wind speeds are more likely to be higher-turbulence, unstable conditions, whereas higher wind speeds tend to be low-turbulence stable conditions. The black line of Fig. 7d will be used to describe the typical TI in the FLORIS model and will be discussed again later.




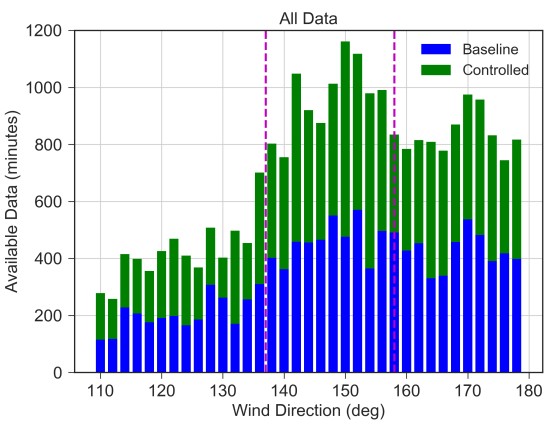

(a) Wind direction distribution

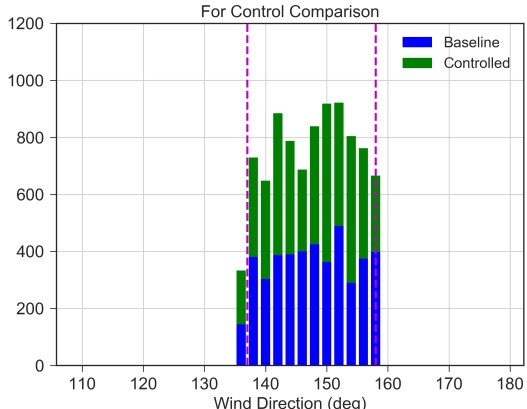

(b) Wind direction distribution of the controlled region and all signals passing quality control

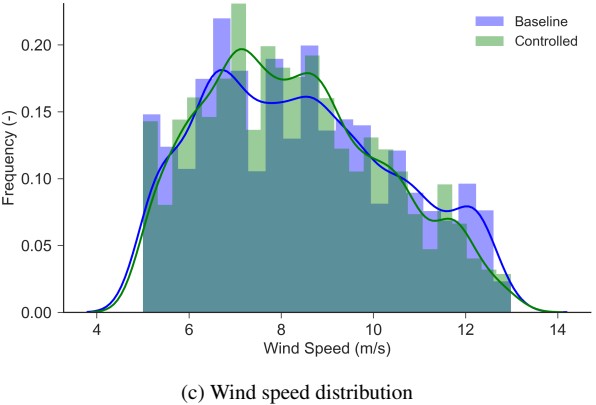

(c) Wind speed distribution

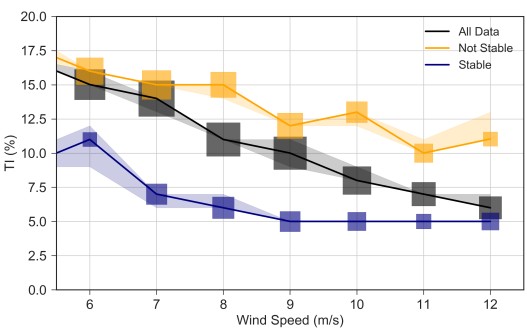

(d) Turbulence intensity by wind speed and stability

**Figure 7.** Atmospheric conditions observed by the sodar at hub-height over the duration of the test campaign.



# 5 Controller Assessment

We first analyze the phase 1 data by considering the performance of the controller in terms of its ability to produce a specific offset by wind speed and wind direction. The exact function of the turbine yaw controller is not known. Therefore, it was difficult to know in advance how effective the method shown in Fig. 5 would be in delivering the desired offsets.

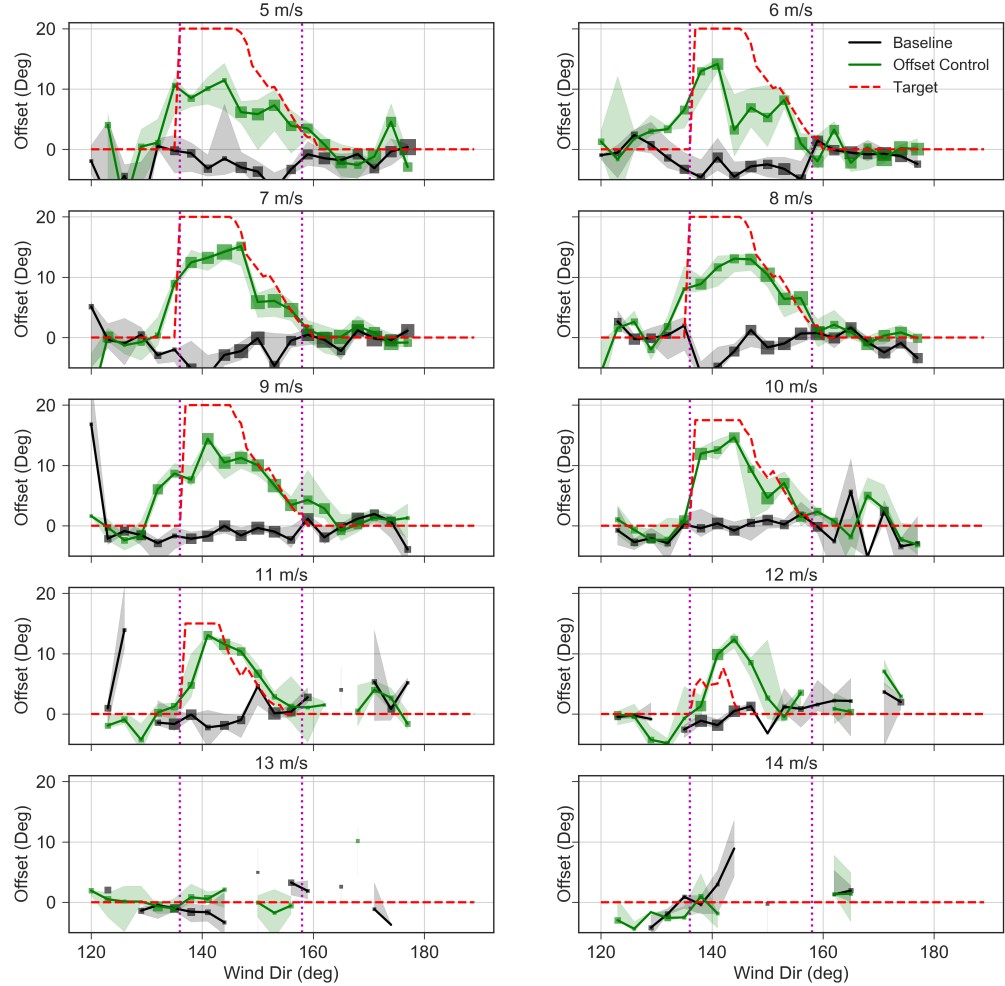

**Figure 8.** Comparison of observed yaw offset (green line) versus target (dashed red line) segregated by wind speeds. Note the achieved offset is computed as the difference between the sodar-measured wind direction and the turbine nacelle heading.




Fig. 8 assesses the performance of this offset controller. Binned by wind speed, the figure shows the median maintained offset. The offset here is computed by comparing the nacelle position of T4 with the measurement of wind direction recorded by the south sodar. Generally, the offsets are reasonably well achieved; however, there is a tendency toward undershoot. The undershoot could be an artifact of temporal averaging over periods with and without offset, which biases computed offsets

5 toward zero. However, we suspect that the undershoot is actually occurring because of the fact that the actual controller is tracking an offset that is zero for most directions, except for a small band about the main waking direction. As the wind speed and direction drift in and out of controlled areas, the averaging effect biases the offset toward zero. This bias could be accounted for in future controller design.

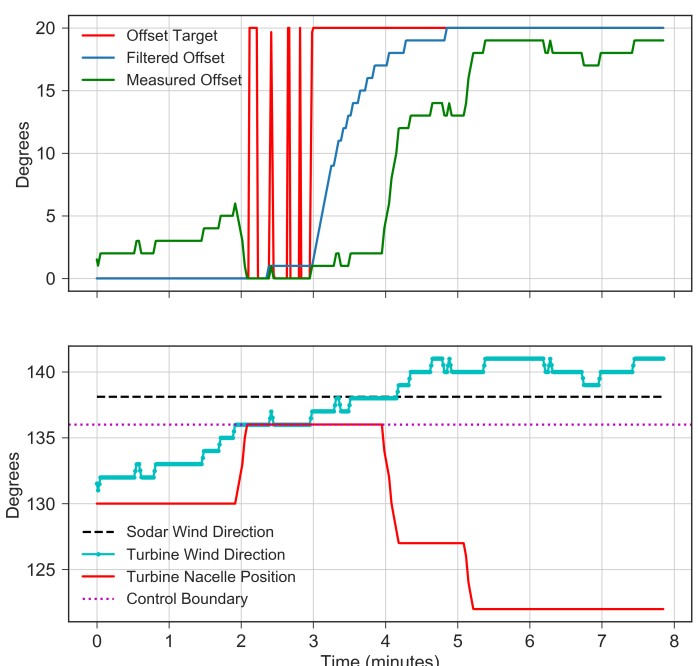

**Figure 9.** Time series data of T4 operation showing the sources of lag in yaw offset control. The figure shows that following the wind direction entering the range at which a 20° offset would be optimal at the 2-minute mark, the offset is not actually achieved until 3 minutes later.

In addition to a steady bias toward smaller-than-targeted offsets, we also noticed a dynamic issue in the controller design. In

10 the initial design phase, it was assumed that, to avoid excessive yawing behaviors, we should both low-pass filter the wind speed and wind direction inputs to the look-up table, as well as the resultant offset sent to the yaw controller. We did not account for the fact that the yaw controller itself acts as a lag filter between changes in wind direction and changes in nacelle yaw position,





and so the yaw offset control system as a whole is probably too slow and a general tendency toward overlagging changes in wind direction was observed. This is illustrated in Fig. 9. At approximately 2 minutes, the wind direction crosses into the region in which a 20-degree offset would be dictated by the static optimal look-up table. The low-pass filtering, however, causes the offset target to lag until the third minute to reach 20°. Then, the filtering of the offset achieves 20° around the fourth minute.

Further, the turbine is observed to begin yawing around the fourth minute, and completes this action in the fifth minute, 3 minutes after the offset could have been optimally applied. Some lag is unavoidable, and potentially desirable to avoid adding two much additional yaw activity to the turbine, but based on this result, the controllers used in the upcoming phase 2 will be designed dynamically, and the filter constants adjusted to account for this.

## 6   Challenges in This Campaign

This phase 1 campaign revealed several challenges that could inform future campaigns, including our future phases. Despite these challenges, the wake steering controller did produce the desired result of increasing power at the downstream turbine.

A first set of challenges corresponds to the site conditions for the south campaign. The topography is complex, but the version of FLORIS used in the design and analysis has no terrain modeling capabilities. This mismatch between the modeling assumptions and reality imparts a degree of uncertainty. Second, T4 and T3 are spaced such that T3 is in the near-wake region

of T4. The version of FLORIS used does not contain a well-tuned near-wake model. Finally, as shown in Fig. 6, the collected data are only about one-third composed of stable atmospheric conditions because of the summer season and longer days. Stable, low-turbulence conditions would be more favorable to wake steering and may occur more frequently in the winter season.

A second set of challenges arise from more practical considerations. Specifically, the only sensor that could be used for the south experiment to measure the inflow is the south sodar, as it is the only one to the south of T4. The south sodar, in

comparison to other instruments, was shown to measure the inflow well; however, it delivers data only once every 10 minutes, and this frequency is too coarse for the data analysis, because 10 minutes will include a diverse mix of wind directions and offsets. The turbine data are delivered at a frequency of 1 Hz, and through trial and error, a compromise of down-sampling the turbine data to 1 minute periods, while up-sampling the sodar data was selected (this up-sampling is done through a "zero-order hold," wherein the data for each minute bin is assigned the 10-minute average). However, in the upcoming north campaign,

more frequent data are available from the lidar and can be used.

Finally, the controller design, wherein we influence the yaw controller without fully understanding its behavior, is a major challenge. If the yaw controller could be directly modified, the delays shown in Fig. 9 could be reduced and performance improved.

## 7   Results

The first step in performing the analysis was selecting a reference for comparing the power of T3 and T4. In previous work, e.g., Fleming et al. (2017b), a reference uncontrolled turbine was preferred as it would provide a reference power that includes





the effects not only of wind speed, but also shear, veer, and TI. For this reason, both T1 and T5 were considered, but a significant amount of noise/noncorrelation was observed likely because both turbines are far and downstream from T4. Ideally, the reference would be parallel to the upstream turbine on a line perpendicular to the controlled wind directions. In addition, the complex terrain varies from T1 to T5.

For this reason, a synthetic reference turbine power was used, based on the measurements of the south sodar. The wind speeds at heights corresponding to the turbine rotor were collected and applied to a weighted average, wherein the weights were proportional to the sector of rotor area the heights correspond with, similar to a rotor-effective wind speed calculation(Wagner et al. (2014)). The hypothetical power of a reference turbine could then be computed using:

$$P_{sodar} = 0.5 \rho A C_p U_{sodar}^3 \tag{3}$$

where $C_p$ is derived from the $C_p$ look-up table included in FLORIS and $\rho$ is the average observed density. This estimated sodar power was then compared with the measured power of T4, when the turbine is not intentionally operating in the offset condition (Fig. 10). The plot shows that correspondence is very close except for near rated.

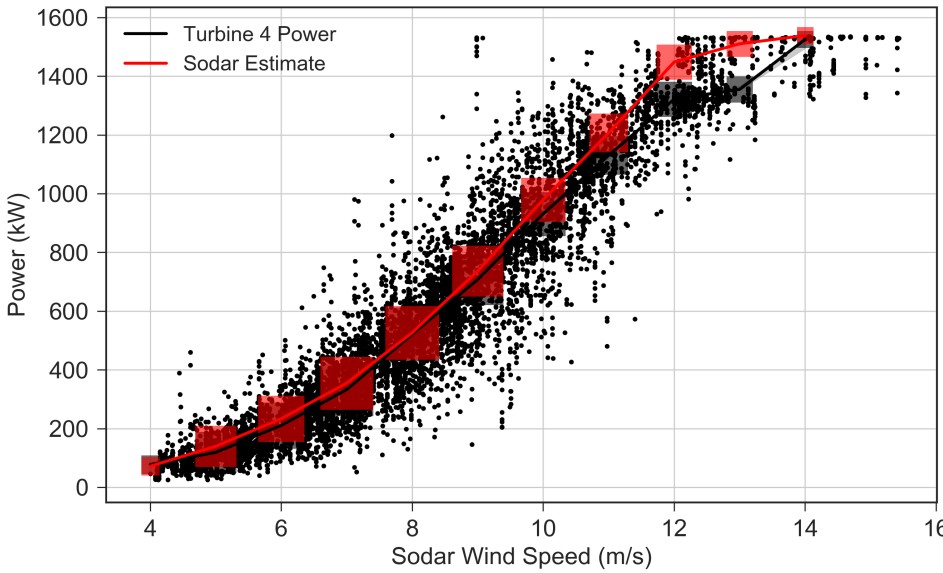

**Figure 10.** Sodar available power fit. The data from T4 are shown in black and the sodar estimate is shown in red. The red boxes refer to the wind speed bins and the size refers to the amount of data in each bin.

To analyze the effect of the wake steering implementation on the control and downstream turbine, the following method of analysis is used. First, the data are limited to include only periods in which both turbines were operating normally, and the

quality of the sodar estimate was above a certain threshold, using quality flags reported by the sodar at each range. Next, all





the data, including the power of T3 and the sodar reference power, are binned into wind direction bins every 2° and according to whether the wake steering controller was toggled on or off.

Then for each bin, an energy ratio is computed, which involves summing all the power measurements of the test turbine, i.e., T3, and the reference turbine, i.e., sodar estimate, and then taking a ratio.

$$R_{Energy} = \frac{\sum_{i=1}^{N} P_i^{Test}}{\sum_{i=1}^{N} P_i^{Ref}} \tag{4}$$

Note that this method is different from a power ratio method in which a power ratio is computed for each set of points and then averaged.

$$R_{Power} = \frac{1}{N} \sum_{i=1}^{N} \left( \frac{P_i^{Test}}{P_i^{Ref}} \right) \tag{5}$$

It is also different than the slope method used in Fleming et al. (2017b).

$$R_{Slope} : \min_{R_{Slope}} ||\boldsymbol{P_{Test}} - R_{Slope} \boldsymbol{P_{Ref}}||_2 \tag{6}$$

where $R_{[...]}$ is the ratio computed through the different methods, $\boldsymbol{P_{Test}}$ and $\boldsymbol{P_{Ref}}$ are vectors of all observed powers for the reference and test turbines, $P_i^{Ref}$ is a single-minute average, and $N$ is the total number of points in a given wind direction bin.

The energy ratio (4) is used for a few reasons. First, changes in relative energy production are more directly related to changes in revenue. Second, the power ratio is an average of ratios instead of the ratio of averages proposed in the energy ratio (5)). The power ratio is more sensitive to small changes in power at low wind speeds, which do not contribute meaningfully to changes in energy production, which is the ultimate goal. The slope method (6) of Fleming et al. (2017b) was able to achieve a weighting of higher wind speeds through slope fitting. However, the energy ratio was finally thought to be more directly related to annual energy production, the overall quantity of interest. The energy ratio represents the increase or decrease in energy produced for a specific wind direction bin.

In addition to computing a single energy ratio for each bin, the process is boot-strapped, in which the data are randomly sampled with replacement and the energy ratio recomputed 1000 times or more depending on the amount of data. The results of these bootstrap iterations are then used to compute 95% confidence intervals. The process is repeated within several differently-defined FLORIS models of the site to provide a point of comparison. The specific set of wind speeds observed within each bin are simulated in four separate FLORIS models (see Table 1). For each of the FLORIS models, and for each 1-minute wind speed and direction observed in the field, a matching FLORIS simulation was run and the power of T3 and T4 tabulated. Note the input TI for FLORIS is set according to the average behavior observed in Fig. 7d.

The energy ratios can be computed from these FLORIS models. The comparison of the aligned and optimal case should present an upper bound on performance if exact offsets and alignments held, whereas the Baseline and Controlled cases show what we expect from this data set.



**Table 1.** FLORIS MODEL DEFINITIONS FOR COMPARISON. Columns yaw 4 and yaw 3 determine what sets the yaw angle for each turbine (aligned being 0° always), optimal being the true optimal, and "From Baseline" means applying the observed yaw angles from the baseline data of the field campaign. Wind conditions are similarly defined.

| Model | Yaw 4 | Yaw 3 | Wind Conditions |
|---|---|---|---|
| **Aligned** | Aligned | Aligned | From Baseline Data |
| **Baseline** | From Baseline | From Baseline | From Baseline Data |
| **Optimal** | Optimal | Aligned | From Controlled Data |
| **Controlled** | From Controlled | From Controlled | From Controlled Data |

Note that all wind speeds are used in this calculation, including those (greater than 12 m/s) in which the 0° offset is actually targeted, even in the controller on mode. With the 10-min sodar rate, and the lag of the controller, it is difficult to draw an exact line in which the controller stops impacting the individual turbine yaw controller. Including all wind speeds also corresponds to the final change in energy. All wind speeds are applied in FLORIS, so their effects are accounted for.

## 7.1 Turbine 3 Analysis

Fig. 11 shows the energy ratios by wind direction for T3. The wake loss is deeper than FLORIS expects (T3 is producing less than 40% of the energy of the reference at nadir); however, as mentioned, this is a difficulty of current near-wake models and the subject of active research. Still, the gain in energy production in the wind direction where the controller is active is observed.

It is useful to note the places in Fig. 11 where the FLORIS model results are somewhat jagged. Fig. 7a shows limited data below 138°. Even without any yaw offsets being applied, as would be the case in this range, the energy ratios can still vary based on the composition of wind speeds (e.g., very high wind speeds have high power ratios, whereas the lowest power ratios occur in the middle of region 2 near 8 m/s.) Given enough data, these effects should wash out, but we can see FLORIS shows a dip in the controlled case at 125° that is indeed observed in the field data. Fortunately, in the controlled band between 136° and 158° , the FLORIS results show less variability, which suggests adequate data collection.

The difference between the Baseline and Controlled cases for both the field data and FLORIS quantifies the impact of yaw control on power production (Fig. 12). In the 10-degree sector between 140° and 150° the average gain in energy is 13%. Compared to FLORIS' predictions, field results show slightly less increased production of energy than the optimal case for the deepest wake region closer to 140°; however, accounting for the undershoot in yaw offsets achieved seems to explain most of that discrepancy. In the right half of the controlled region, however, we note a tendency to exceed both the controlled and optimal FLORIS predictions for energy gain. The underperformance on the left of the control region in Fig. 12 near 140° could be the problem of yaw offset undershoot, and therefore a focus of controller design in future work. The overperformance on the right side may indicate the need for better modeling of wake steering partial wake and could be improved upon using





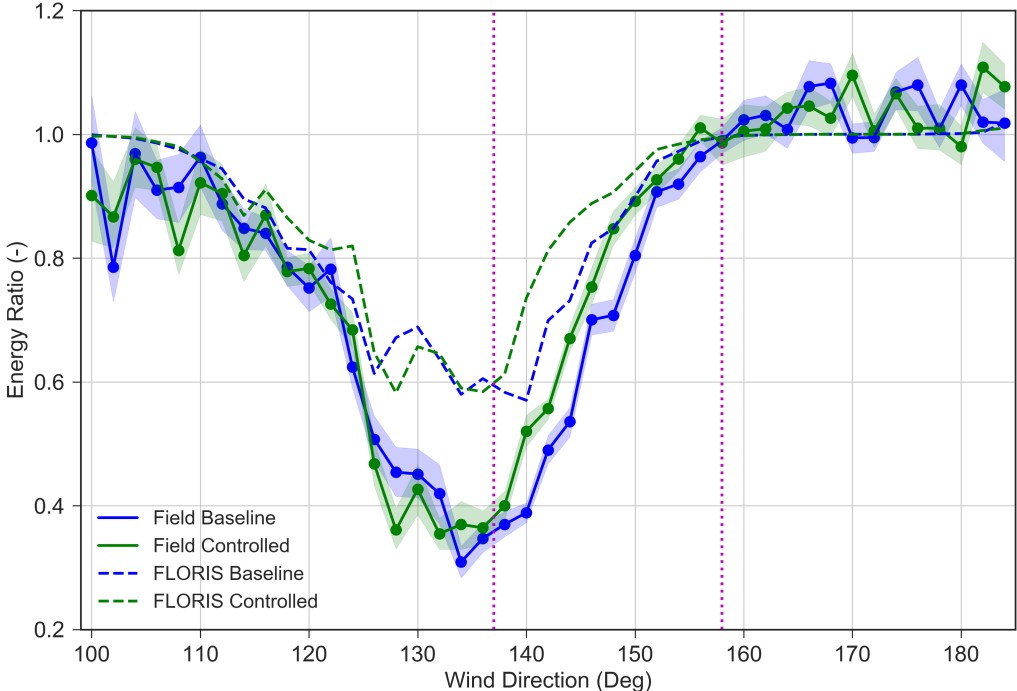

**Figure 11.** Energy ratio for T3 for field data and the Baseline and Controlled FLORIS cases (see Table 1). An energy ratio of 0.5 corresponds to a production of 50% of the total expected based on the measured inflow without considering wakes. The vertical magenta lines indicate the region where control is applied and a difference between the Baseline and Controlled is expected.

newer vortex-based curl models in FLORIS, such as identified in Fleming et al. (2018) and Martínez-Tossas et al. (2015). On the other hand, another possibility is that uncertainty in wind direction lowers the performance on the left and raises it on the right through averaging. Finally, the data outside of the control bands, while noisy, indicate an average around 0, underlying the significance of the non-zero apparent average in the control region.

5     The difference in energy production is more clearly realized in stable conditions as shown in Fig. 13, which segregates stable conditions from not-stable conditions. This distinct improvement in stable conditions could be because wake steering is more effective in stable conditions. Additionally, in stable conditions, atmospheric inflow is more homogeneous and therefore easier to measure. Upcoming winter measurements from the north for phase 2 will consist of more stable measurements due to the longer nights and may shed more light on the role of atmospheric stability in wake steering.




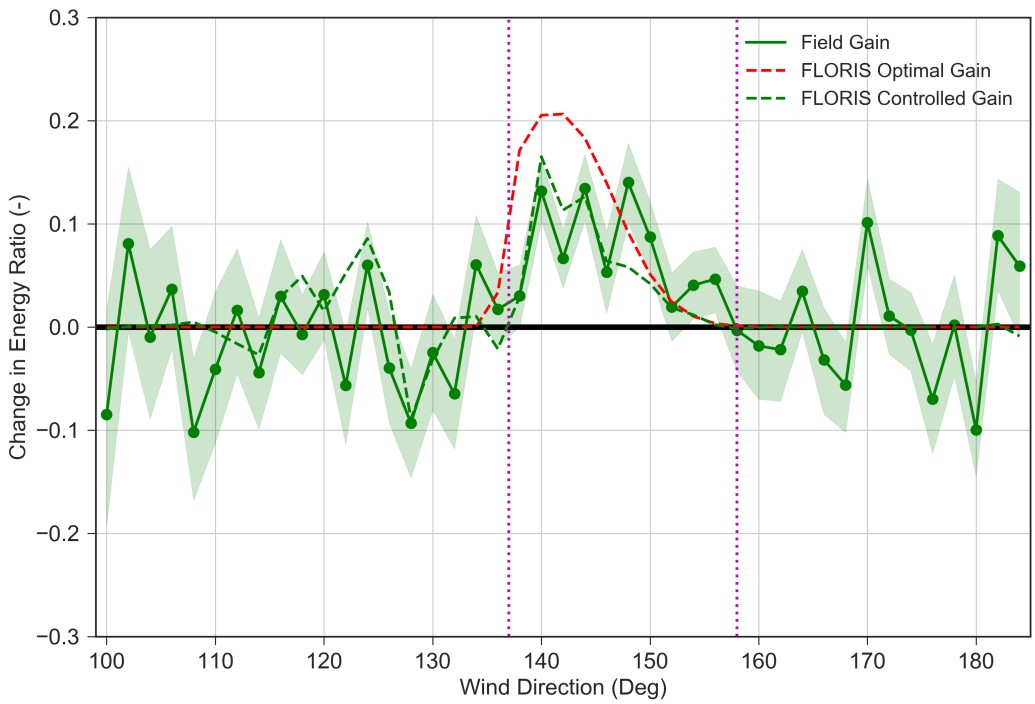

**Figure 12.** Change in the energy ratio of T3 from field data. Expectations from FLORIS using the actually achieved controlled offsets (Controlled - Baseline) and optimal (Optimal - Aligned) are shown to indicate comparison with expectations given control values, and what is optimally expected.

## 7.2 Aggregate Analysis

Finally, the previously mentioned analysis is repeated, but using the aggregated power of T4 and T3, so that the losses in energy coming from offseting the yaw of T4 are deducted from the gains made downstream (Fig. 14 showing the energy ratios and Fig. 15 shows the difference betwen them). Again, we gain less energy than FLORIS expects in the left half and more than expected on the right half. Intriguingly, but perhaps coincidentally, if we compute the energy gain over the band of direction for which the controller is active, both *optimal* FLORIS and the field data yield a 3.7% increase in energy, in which case the under and overperformance balance each other out.

## 8 Conclusions

We present the initial results from a first phase of a field campaign evaluating wake steering at a commercial wind farm. For two closely spaced turbines, an approximate 13% increase in energy was measured on the downstream turbine over a 10° sector.

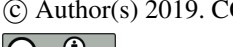


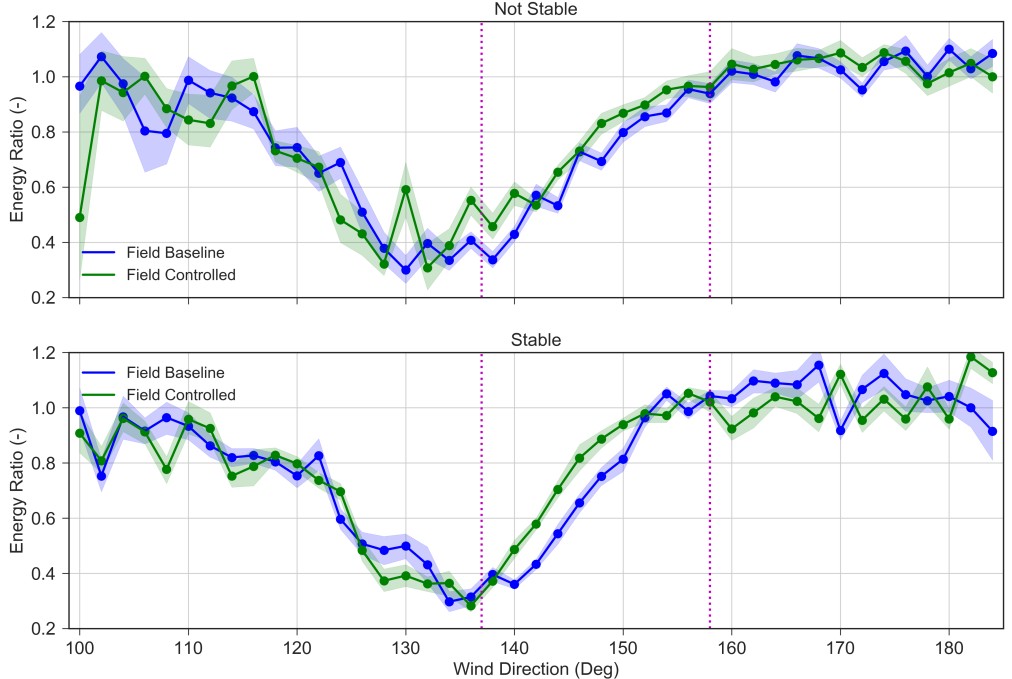

**Figure 13.** Energy ratio of T3, as in Fig. 11, divided into stable and not-stable conditions.

The gains in energy were compared to predictions made using the FLORIS model used to design the applied controllers. The overall gains were consistent with predictions from FLORIS; however, this agreement was due to less-than-expected gains in full-wake conditions, and more-than-expected gains in partial wakes.

This initial stage of the wake steering campaign identified several areas for improvement in future work, such as aspects

5  of dynamic controller design, time filtering, and uncertainty quantification. Difficulties with this particular south campaign, including complex terrain and summer atmospheric conditions, were identified as possible sources of improvement as the campaign moves to northern winter conditions. Additionally, near-wake modeling presented a challenge in accurately modeling the wake losses, and therefore may have realized a less-than-optimal controller. Still, the overall gains in energy were in line with prior expectations from FLORIS.

10  All together, the authors hope that the results presented might therefore represent a baseline for possibility of gains from wake steering. Better modeling and controls, simpler site conditions, and the exploitation of vortex modeling and larger arrays of turbines all present hopeful avenues for continued improvements. An upcoming companion paper on the second phase of the experiment will review results including opportunities for improvement identified in this paper.




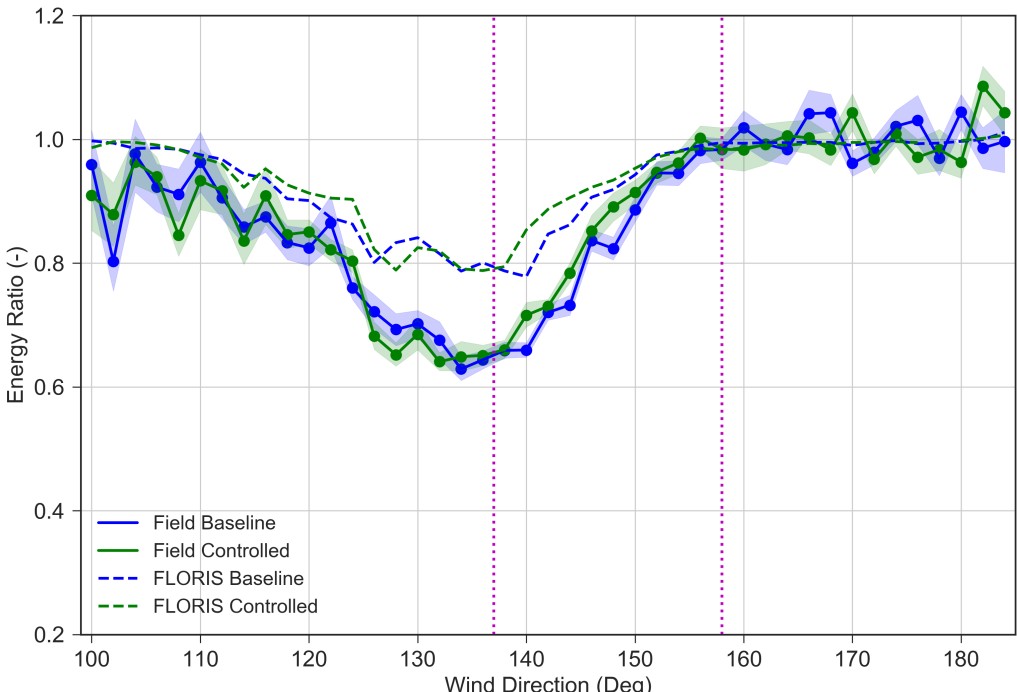

**Figure 14.** Energy ratio of the summed energy of T3 and T4.

*Acknowledgements.* This work was authored [in part] by the National Renewable Energy Laboratory, operated by Alliance for Sustainable Energy, LLC, for the U.S. Department of Energy (DOE) under Contract No. DE-AC36-08GO28308. Funding provided by the U.S. Department of Energy Office of Energy Efficiency and Renewable Energy Wind Energy Technologies Office. The views expressed in the article do not necessarily represent the views of the DOE or the U.S. Government. The U.S. Government retains and the publisher, by accepting the

5  article for publication, acknowledges that the U.S. Government retains a nonexclusive, paid-up, irrevocable, worldwide license to publish or reproduce the published form of this work, or allow others to do so, for U.S. Government purposes.




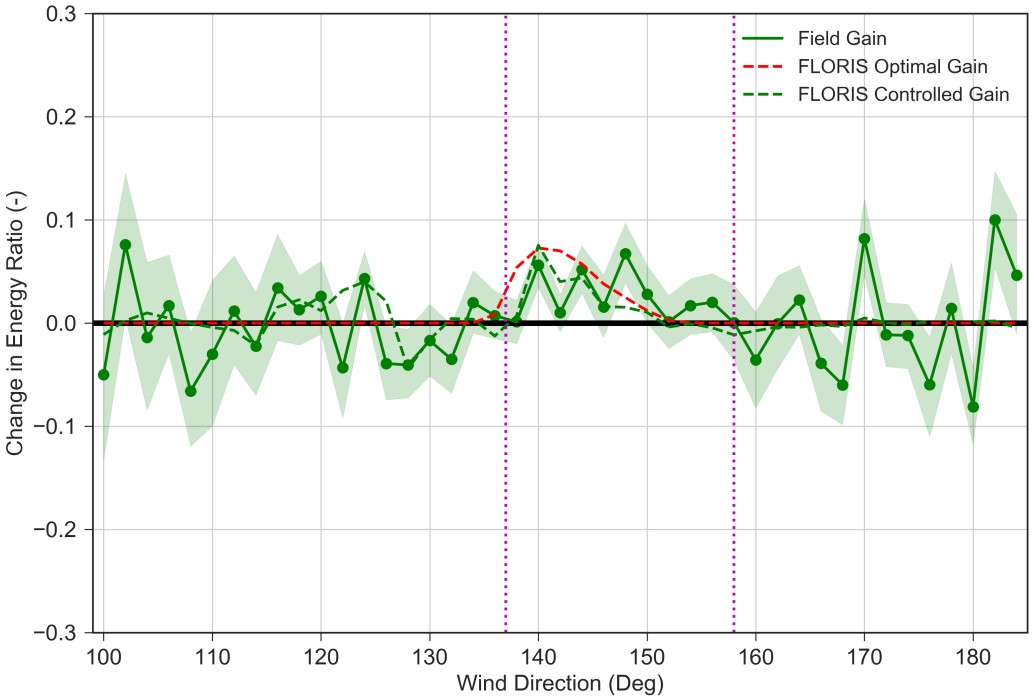

**Figure 15.** Combined change in energy ratio for T3 and T4.

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
