# Peer review of "Initial Results From a Field Campaign of Wake Steering Applied at a Commercial Wind Farm: Part 1"

_Wind Energy Science, 2019_

## Referee Comment (RC1) · Anonymous Referee #1 · 27 Feb 2019

Very good paper, and very clear. Could be accepted as is, but just one or two very minor points which could be improved in any revised version: Page 2 line 17, spelling of "University" Page 15, last sentence is not entirely clear and a bit confusing. Table 1: the difference in wind conditions between baseline and controlled could also be explained more clearly.

---

## Short Comment (SC1) · 21 Mar 2019

The paper clearly describes a field campaign illustrating the potential for wake steering in an actual operational wind farm. Very interesting work that is valuable to the community.

A small comment: the current south campaign focuses on the T3 and T4 turbines, which are spaced exceptionally close together (<3 rotor diameters). It would be nice if the authors could comment on how they expect results on e.g. power gain to generalize to more common spacings (let's say >7D).

The effect of this larger spacing could be twofold: on the one hand, wake losses will be lower but on the other hand, the wake centerline should be displaced over a larger distance (see, e.g. Bastankhah & Porte-Agel, Experimental and theoretical study of wind turbine wakes in yawed conditions, J Fluid Mech 2016).

Very nice work overall.

---

## Referee Comment (RC2) · Anonymous Referee #2 · 10 Apr 2019

The paper presents the initial results from a wake steering field campaign. Overall, the paper is very well-written and includes very important findings. Lessons learned are also presented which are very helpful for the community. Although the quality of the paper is already very high and in principle could be published as submitted, the minor issues mentioned below might be helpful to further improve the paper.

Author list: depending on their contribution you might consider to move some persons to the acknowledgment section.

Abstract: You could consider to add the 3.7% of the aggregated analysis as well. Wake steering is a wind farm control method and thus the effect to the combined energy is

also very important.

Figure 4: You could add some other wind speed as well to provide a better feeling for the 2 D lookup-table.

Text to Figure 5: You could provide the time constants of the low pass filter. It is partly visible from Fig. 9, but might help to better understand the issue with the time offset. And why did you use a filter for the offset? Wouldn't it be enough to have the inputs filtered?

Equation 1 and 2: You could mention that u* is the frictional velocity, and ′ means perturbations. And isn't the unit m missing in L<-1000?

Figure 6 and 7c are without a box, all others have one.

Figure 7c: After some time, I figured out that the third color is due to the overlay of baseline and controlled case. Couldn't you simply have to column for each wind speed bin? And what are the lines?

Figure 7d and 8: What does the shading here mean? The 95% confidence interval seems to be unlikely, since the mean is not centered.

Equation 3: You could consider to describe all components of the equation. It is pretty clear what A means, but just for the sake of completeness. And couldn't you use the mean power curve from T4 outside of the controlled region to get the power from the sodar within the controlled region instead of the power coefficient to avoid issues with wind speeds near rated?

Equation 4-6: you could consider to write the subscripts in normal text mode (not math mode).

Page 14, line 15: There is an additional )

Figure 11-15: Is the shading the 95% confidence interval? If so, you could add this to the captions or in the text. Also, the placing is a bit strange. . . e.g. Figure 15 is on page

Interactive
comment
20 and the reference on page 17... but might be the usual latex mystery.

Looking forward to the paper from the north campaign!

---

## Author Comment (AC2) · 26 Apr 2019

Thank you Wim for this comment, I hope it will not be too long for the part 2 of this paper, which will detail the north campaign at the more typical spacing. Sorry to not have an answer just yet, as the analysis is still preliminary while the second phase is ongoing. Thank you very much for your comment!!

---

## Author Response (AR1)

Reviewer Response
April 24, 2019

We thank the reviewers for their helpful and constructive remarks. We have prepared a revised version of the paper including the suggestions of the reviewers, and below outline these changes in response to your recommendations.

Additionally, since the original submission, both the FLORIS model and the energy ratio method have been improved somewhat independently of this research, and these improvements are reflected somewhat in the final analysis. The energy ratio method is described in greater detail, and a link to the source code are now included.

**Reviewer 1:**

*Page 2 line 17, spelling of "University"*

This is fixed

*Page 15, last sentence is not entirely clear and a bit confusing.*

This sentence is removed

*Table 1: the difference in wind conditions between baseline and controlled could also be explained more clearly.*

More text was included to make these cases more clear:

The energy ratio calculation is repeated on several differently-defined FLORIS models of the site to provide a point of comparison. An ``Aligned" case simulates every observed wind speed and direction in the Baseline field data with all turbines perfectly aligned to the flow, while a ``Baseline" case uses the actual small offsets observed. An "Optimal" case, simulates all the wind speeds and direction in the Controlled field data using the exact offset requested by the control strategy, while ``Controlled" applies the actual achieved offset.
These four settings are summarized in Table~\ref{tab:floris}. For each of the FLORIS models, and for each 1-minute wind speed and direction observed in the field, a matching FLORIS simulation was run and the power of T3 and T4 tabulated and the energy ratio computed. When considering gains in energy, ``FLORIS Optimal Gain" refers to the change from ``Aligned" to ``Optimal" whereas "FLORIS Controlled Gain" refers to the change from "Baseline" to "Controlled".

**Reviewer 2:**

*Author list: depending on their contribution you might consider to move some persons to the acknowledgment section.*

The test campaign was a large group undertaking and it was decided to share authorship among those who worked across aspects of the campaign

Abstract: You could consider to add the 3.7% of the aggregated analysis as well. Wake steering is a wind farm control method and thus the effect to the combined energy is also very important.

Good point, this value is now added

Figure 4: You could add some other wind speed as well to provide a better feeling for the 2 D lookup-table.

This figure is updated to include additional wind speeds and the caption updated to reflect

Text to Figure 5: You could provide the time constants of the low pass filter. It is partly visible from Fig. 9, but might help to better understand the issue with the time offset.

Each of the filters has a 30-s time constant this is now included in the description.

And why did you use a filter for the offset? Wouldn't it be enough to have the inputs filtered?

It was felt more cautious to filter the offset to smooth the step changes when the wind direction changes from 139-138, however, indeed this appears now unnecessary and is not used in phase 2

Equation 1 and 2: You could mention that u* is the frictional velocity, and 0 means perturbations.

This is corrected

And isn't the unit m missing in L<-1000?

This is corrected

Figure 6 and 7c are without a box, all others have one.

Figure 7c was updated to include a box to match the other subplots in that figure, while figure 6 was left for now since it stands alone, however this can be added in the final version

Figure 7c: After some time, I figured out that the third color is due to the overlay of baseline and controlled case. Couldn't you simply have to column for each wind speed bin? And what are the lines?

Figure 7c is replaced by a dodged histogram to make the comparison more visually apparent

Figure 7d and 8: What does the shading here mean? The 95% confidence interval seems to be unlikely, since the mean is not centered.

The points are medians and the shadings are 95% confidence of the medians, descriptions are updated in the legends and text

Equation 3: You could consider to describe all components of the equation. It is pretty clear what A means, but just for the sake of completeness.

These descriptions are added

 And couldn't you use the mean power curve from T4 outside of the controlled region to get the power from the sodar within the controlled region instead of the power coefficient to avoid issues with wind speeds near rated?

The complexity of the terrain as the wind moves to the west or east cause the relationship between sodar and turbine to be non-constant outside of the controlled region unfortunately

Equation 4-6: you could consider to write the subscripts in normal text mode (not math mode).

This change is made

Page 14, line 15: There is an additional )

Fixed

 Figure 11-15: Is the shading the 95% confidence interval? If so, you could add this to the captions or in the text.

This is the 95% interval, and the description now included the caption of figure 11:

Figure 11.Energy ratio for T3 for field data and the Baseline and Controlled FLORIS cases (see Table 1). An energy ratio of 0.5 corresponds to a production of 50% of the total expected based on the measured inflow without considering wakes. The vertical magenta lines indicate the region where control is applied and a difference between the Baseline and Controlled is expected. Size of circles at each point indicate the number of points in the bin, while the bands indicate 95% confidence as computed by the bootstrapping.

Also, the placing is a bit strange. . . e.g. Figure 15 is on page 20 and the reference on page 17... but might be the usual latex mystery

This is indeed a latex mystery, but we hope that working with the journal editors, the placement of figures can be more reasonably situated in the final version,

[revised manuscript text omitted]

To analyze the effect of the wake steering implementation on the control and downstream turbine, the following method of
20  analysis is used. First, the data are limited to include only periods in which both turbines were operating normally, and the quality of the sodar estimate was above a certain threshold, using quality flags reported by the sodar at each range. Next, all the data, including the power of T3 and the sodar reference power, are binned into wind direction bins every 2° (with 1° of overlap between adjacent bins as this was found useful in clarifying trends in the available data) and according to whether the wake steering controller was toggled (Controlled) or off (Baseline).
25  Then for each bin, an energy ratio is computed, which involves  a weighted summation of all the power measurements $P^{Test}$ of the test turbine (the determination of the weights $w$ will be detailed later), i.e., T3, and the reference turbine, i.e., sodar estimate $P^{Ref}$, and then taking a ratio of the two.

$$R_{Energy} = \frac{\sum_{i=1}^{N} w_i P_i^{\text{Test}}}{\sum_{i=1}^{N} w_i P_i^{\text{Ref}}} \tag{4}$$

[Figure]

**Figure 10.** Sodar available power fit. The data from T4 are shown in black and the sodar estimate is shown in red. The red boxes refer to the wind speed bins and the size refers to the amount of data in each bin.

Note that this method is different from a power ratio method in which a power ratio is computed for each set of points and then averaged.

$$R_{Power} = \frac{1}{N} \sum_{i=1}^{N} \left( \frac{P_i^{Test}}{P_i^{Ref}} \right) \tag{5}$$

It is also different than the slope method used in Fleming et al. (2017b).

5   $R_{Slope} : \min_{R_{Slope}} ||\boldsymbol{P}_{\text{Test}} - R_{Slope} \boldsymbol{P}_{\text{Ref}}||_2 \tag{6}$

where $R_{[...]}$ is the ratio computed through the different methods, $\boldsymbol{P_{Test}}$ and $\boldsymbol{P_{Ref}}$ are vectors of all observed powers for the reference and test turbines, $P_i^{Ref}$ is a single-minute average, and $N$ is the total number of points in a given wind direction bin.

The energy ratio (4) is used for a few reasons. First, changes in relative energy production are more directly related to changes in revenue. Second, the power ratio is an average of ratios instead of the ratio of averages proposed in the energy ratio

10   (5). The power ratio is more sensitive to small changes in power at low wind speeds, which do not contribute meaningfully to changes in energy production, which is the ultimate goal. The slope method (6) of Fleming et al. (2017b) was able to achieve a weighting of higher wind speeds through slope fitting. However, the energy ratio was finally thought to be more directly

related to annual energy production, the overall quantity of interest. The energy ratio represents the increase or decrease in energy produced for a specific wind direction bin.

In computing the energy ratio, a wind-speed-based weighting strategy helps to more quickly converge the analysis and reduce changes in energy ratios due to variations of the compositions of wind speeds for the two controllers within each wind direction bin with respect to differences owed to changes in controller. The main idea is that for each wind direction, the power values collected are binned according to wind speed (as measured by the sodar) and the total energy for the baseline and controlled cases are the weighted sum of the powers, where the weight is the number of points the other control setting has in this wind speed bin, out of the total. For example, if there are 10 samples of Baseline taken at 13 m/s, and 5 of Controlled, the baseline points are weighted by 1/3, and the controlled by 2/3, so the final energy ratios to be compared are more approximately even in terms of wind speed distributions represented. If the bins are perfectly balanced, there are an equivalent number of Baseline and Controlled points at each wind speed, the weights have no effect.

In addition to computing a single energy ratio for each bin, the process is boot-strapped, in which the data are randomly sampled with replacement and the energy ratio recomputed 1000 times or more depending on the amount of data. The results of these bootstrap iterations are then used to compute 95% confidence intervals. , and these are indicated in the plots of energy ratio as semi-transparant bands.

The method of computing energy ratios is now included within the open-source FLORIS model https://floris.readthedocs.io.

Note that all wind speeds are used in this calculation, including those (greater than 12 m/s) in which the 0° offset is actually targeted, even in the controller on mode. With the 10-min sodar rate, and the lag of the controller, it is difficult to draw an exact line in which the controller stops impacting the individual turbine yaw controller. Including all wind speeds also corresponds to the final change in energy.

The energy ratio calculation is repeated on several differently-defined FLORIS models of the site to provide a point of comparison.  An "Aligned" case simulates every observed wind speed and direction in the Baseline field data with all turbines perfectly aligned to the flow, while a "Baseline" case uses the actual small offsets observed. An "Optimal" case, simulates all the wind speeds and direction in the Controlled field data using the exact offset requested by the control strategy, while "Controlled" applies the actual achieved offset. These four settings are summarized in Table 1. For each of the FLORIS models, and for each 1-minute wind speed and direction observed in the field, a matching FLORIS simulation was run and the power of T3 and T4 tabulated  and the energy ratio computed. When considering gains in energy, "FLORIS Optimal Gain" refers to the change from "Aligned" to "Optimal" whereas "FLORIS Controlled Gain" refers to the change from "Baseline" to "Controlled".

[revised manuscript text omitted]